# Exosome-Based Cell Homing and Angiogenic Differentiation for Dental Pulp Regeneration

**DOI:** 10.3390/ijms24010466

**Published:** 2022-12-27

**Authors:** Venkateswaran Ganesh, Dongrim Seol, Piedad C. Gomez-Contreras, Henry L. Keen, Kyungsup Shin, James A. Martin

**Affiliations:** 1Department of Orthopedics and Rehabilitation, University of Iowa, Iowa City, IA 52242, USA; 2Department of Roy J. Carver Biomedical Engineering, University of Iowa, Iowa City, IA 52242, USA; 3Department of Orthodontics, University of Iowa, Iowa City, IA 52242, USA; 4Iowa Institute of Human Genetics, University of Iowa, Iowa City, IA 52242, USA

**Keywords:** pulp regeneration, dental pulp stem cells, exosomes, angiogenesis, cell homing, microRNA profile

## Abstract

Exosomes have attracted attention due to their ability to promote intercellular communication leading to enhanced cell recruitment, lineage-specific differentiation, and tissue regeneration. The object of this study was to determine the effect of exosomes on cell homing and angiogenic differentiation for pulp regeneration. Exosomes (DPSC-Exos) were isolated from rabbit dental pulp stem cells cultured under a growth (Exo-G) or angiogenic differentiation (Exo-A) condition. The characterization of exosomes was confirmed by nanoparticle tracking analysis and an antibody array. DPSC-Exos significantly promoted cell proliferation and migration when treated with 5 × 10^8^/mL exosomes. In gene expression analysis, DPSC-Exos enhanced the expression of angiogenic markers including vascular endothelial growth factor A (VEGFA), Fms-related tyrosine kinase 1 (FLT1), and platelet and endothelial cell adhesion molecule 1 (PECAM1). Moreover, we identified key exosomal microRNAs in Exo-A for cell homing and angiogenesis. In conclusion, the exosome-based cell homing and angiogenic differentiation strategy has significant therapeutic potential for pulp regeneration.

## 1. Introduction

Dental pulp is a highly vascularized soft tissue with a layer of odontoblasts at the inner dentin surface. One of the main physiological functions of dental pulp is to provide nutrition to dentin and detect unhealthy stimuli as a biosensor. Current endodontic therapy for dental caries, which is one of the most prevalent infectious diseases in the world [1], is a procedure for replacing vital pulp with synthetic materials via root canal therapy (RCT). Calcium hydroxide paste (CHP) and mineral trioxide aggregate (MTA) have been commonly used as pulp-capping materials due to their antibacterial properties, biocompatibility, and reparative dentin bridge formation [2,3,4,5]. However, CHP has several disadvantages for long-term use such as bacterial leakage into the dental pulp, poor cohesive strength, and high solubility [6]. Similarly, the drawbacks of MTA are its expensive cost, discoloration of the tooth, and long setting time [7]. Moreover, RCT-treated pulpless teeth can lose their ability to sense environmental changes and maintain dentin regeneration, ultimately compromising the mechanical integrity of the teeth. As an alternative, vital pulp therapy (VPT), which is defined as a restorative dental treatment that aims to preserve and maintain pulp tissue, is beneficial for young patients who have a high healing capacity for pulp regeneration [8,9]. Recently, the potential for successful VPT and pulp regeneration is increasing due to the use of mesenchymal stem cells (MSCs) that can differentiate into specialized cells [10,11,12]. However, the transplantation of MSCs incurs high costs and risks associated with ex vivo cell expansion [13]. Thus, there is an urgent need to develop alternative therapeutic methodologies for pulp regeneration.

A cell homing strategy is an effective approach for pulp regeneration in endodontics, as it can recruit endogenous dental pulp stem cells (DPSCs) residing in the pulp tissue to the damaged site in need of repair [14]. The DPSCs have unlimited self-renewal with high clonogenicity and lineage-specific multipotent abilities [15,16]. These characteristics empower DPSCs to migrate locally to sites of injury where they proliferate and differentiate as needed to replace damaged tissue [17]. Unlike cell transplantation, this DPSC homing approach does not require multiple surgical procedures for cell harvesting and transplantation.

Exosomes have attracted attention due to their great potential to promote intercellular communication leading to enhanced cell recruitment, differentiation to specific cell lineage, and tissue repair [18,19,20,21,22]. Exosomes represent an important mode of intercellular communication as they contain a variety of bioactive molecules including deoxyribonucleic acid (DNA), ribonucleic acid (RNA), lipids, and proteins [23]. In particular, microRNAs (miRNAs) have been increasingly recognized for their therapeutic potential [24]. Recent studies have demonstrated that damaged tissues are repaired by the paracrine signaling of exosomes rather than direct proliferation and differentiation [25]. This paracrine effect implies that exosome therapy has a clinical advantage over stem cell transplantation in terms of immune response and tumorigenesis. In addition, MSCs are promising sources of exosomes. Recently, exosomes isolated from MSCs have shown therapeutic potential in heart, skin, and hyaline cartilage regeneration [18,21,22,26,27]. More importantly, a series of prior studies have shown DPSC-derived exosome (DPSC-Exo) enhanced cell recruitment and angiogenesis [28,29,30]. In particular, a conditioned medium or exosomes cultured under angiogenic differentiation have great potential for pulp regeneration.

We hypothesized that exosomes isolated from undifferentiated or angiogenic-differentiated DPSCs will play an important role in inducing cell migration and angiogenesis of naïve DPSCs, respectively. The objective of this study was to evaluate the effects of DPSC-Exos on in vitro cell homing and angiogenesis for pulp regeneration.

## 2. Results

### 2.1. Characterization of DPSCs and DPSC-Exos

The stem cell characterization of rabbit DPSCs was validated by angiogenic and odontogenic multipotential differentiation. In an endothelial tube formation assay, elongated DPSCs formed capillary-like networks when they were pre-cultured in an angiogenic induction medium (Figure 1B), whereas non-induced cells were clustered together without any tube formation (Figure 1A). In odontogenic differentiation, strong deposits of calcium were observed time-dependently in Alizarin Red S staining (Figure 1C,D).

Exosomes were isolated from DPSCs cultured under a growth (Exo-G) or angiogenic differentiation (Exo-A) condition. The characterization of exosomes was performed by a Nanoparticle Tracking Analyzer (NTA), Exo-Check^TM^ exosome antibody array, scanning electron microscope (SEM), and cellular uptake assay. The concentration of Exo-G (11.6 × 10^10^ particles/mL) was approximately two times higher than that of Exo-A (5.5 × 10^10^ particles/mL), while the mean size of particles was similar (104.3 nm Exo-G and 103.9 nm Exo-A) (Figure 1E). Both Exo-G and Exo-A showed a positive expression of ANXA5, TSG101, FLOT1, ICAM, ALIX, and CD81 as exosome markers, and a negative expression of GM130 as a negative marker (Figure 1F). Under SEM, both exosomes had a spheroid shape approximately 100 nm in size (Figure 1G,H). In an uptake assay, both types of exosomes labeled with PKH67 green fluorescence highly internalized into the DPSCs (Figure 1I,J).

### 2.2. Effect of DPSC-Exos on Cell Toxicity, Proliferation, Migration, and Angiogenic Differentiation

In the cytotoxicity test, vehicle controls (4.3%, 9.0%, and 16.3% (*v*/*v*) phosphate buffered saline (PBS)) did not impact cell viability. Therefore, all data from vehicle controls and no exosome treatment control were pooled together. There was no significant cell death in less than 5 × 10^8^ exosomes. However, the viability was significantly decreased in both 5 × 10^9^ Exo-G (*p* = 0.003) and Exo-A (*p* = 0.003) (Figure 2A). For 4 days, 5 × 10^8^ exosome treatments resulted in a slight increase in cell proliferation in both Exo-G (*p* < 0.001) and Exo-A (*p* < 0.001) (Figure 2B). In a migration assay using Transwell^®^ plates, exosome treatment with a concentration of 5 × 10^8^ exosomes promoted dramatic cell migration in both groups of Exo-G (*p* < 0.001) and Exo-A (*p* < 0.001) (Figure 2C,D). The effect was similar for a positive control which contained 10% (*v*/*v*) exosome-depleted FBS. In contrast, a low concentration of exosomes (5 × 10^7^) did not affect cell migration.

To evaluate the effect of DPSC-Exos on angiogenic differentiation, DPSCs were treated with 5 × 10^8^/mL exosomes in basal or complete induction media (bAM or cAM, respectively) for 7 days. There was no effect of DPSC-Exos on the expression of angiogenic markers when cultured in bAM (Figure 3). On the other hand, angiogenic markers of VEGFA, FLT1 (VEGFR1), and PECAM1 were highly expressed when treated with Exo-G and Exo-A in cAM. Both exosome groups showed statistically higher expressions of VEGFA (Exo-G versus control: *p* < 0.001 and Exo-A versus control: *p* = 0.004), FLT1 (Exo-G versus control: *p* < 0.001 and Exo-A versus control: *p* < 0.001), and PECAM1 (Exo-G versus control: *p* < 0.001 and Exo-A versus control: *p* < 0.001). The degree of fold changes between Exo-G and Exo-A was similar in FLT1 and PECAM1, while Exo-G induced a higher expression of VEGFA (*p* = 0.026 versus Exo-A).

### 2.3. Exosomal miRNA Profile

In NGS analysis, we found a total of 474 mature and 254 hairpin miRNAs in Exo-G and a total of 459 mature and 253 hairpin miRNAs in Exo-A. Figure 4A,B show highly expressed miRNAs with over 10,000 total read counts in Exo-G and Exo-A, respectively. Mature miRNAs with over 10,000 total read counts in both Exo-G and Exo-A were shown in Figure 4C. In particular, four miRNAs including ocu-miR-199a-3p, ocu-miR-221-3p, ocu-miR-24-3p, and ocu-miR-21-5p were detected in over 100,000 total read counts. In contrast, there was a distinct expression of miRNA levels in Exo-A when compared with that in Exo-G (Figure 4D). Exo-A isolated under angiogenic differentiation conditions included 30 up-regulated (ocu-miR-708-5p, ocu-miR-205-5p, ocu-miR-708-3p, etc.) and 32 down-regulated (ocu-miR-146a-5p, ocu-miR-503-5p, ocu-miR-20b-5p, etc.) mature miRNAs (Figure 4E). There were 18 hairpin miRNAs that significantly changed in Exo-A, of which 9 hairpin miRNAs (ocu-miR-708-5p, ocu-miR-205-5p, ocu-miR-708-3p, etc.) increased and 9 hairpin miRNAs (ocu-miR-708-5p, ocu-miR-205-5p, ocu-miR-708-3p, etc.) decreased. All differentially expressed mature and hairpin miRNAs are listed in Table 1 and Table 2, respectively. Exosomal miRNAs with a significant expression of both mature and hairpin miRNAs in Exo-A were ocu-miR-708 (up), ocu-miR-134 (up), ocu-miR-125b (up), ocu-miR-140 (up), ocu-miR-29b (up), ocu-miR-214 (up), ocu-miR-574 (down), ocu-miR-503 (down), ocu-miR-30a (down), ocu-miR-146a (down), and ocu-miR-671 (down). In addition, 88 and 36 novel miRNAs were identified in Exo-G and Exo-A, respectively (Appendix A).

## 3. Discussion

The main objective of this study was to evaluate the therapeutic potential of characterized exosomes to stimulate cell homing and angiogenic differentiation for pulp regeneration. Exosomes were isolated from DPSCs cultured under a growth or angiogenic differentiation condition for Exo-G and Exo-A, respectively. Our results revealed that both DPSC-Exos significantly promoted cell proliferation (Figure 2B), migration (Figure 2C,D), and angiogenic differentiation (Figure 3) when treated with 5 × 10^8^/mL exosomes. In NGS, we profiled miRNAs encapsulated in DPSC-Exos to identify key miRNAs for cell homing and angiogenesis (Figure 4, Table 1 and Table 2).

In this study, DPSC-Exos were isolated by a polymer-based precipitation method which has the advantages of saving time and labor, being easily scalable, and having a higher yield compared to ultracentrifugation [31]. According to a standard guideline from the International Society of Extracellular Vesicles (ISEV) [32], isolated exosomes were characterized in terms of size distribution and concentration, exosome-specific protein markers, morphology, and cellular uptake. In NTA, both DPSC-Exos showed a similar size distribution with a peak at approximately 104 nm, whereas Exo-G had an approximately two-fold enrichment in the concentration compared to Exo-A (Figure 1E). This difference can be explained by different cell types which affect exosome yield. Although there was no direct comparison between DPSCs and angiogenic differentiated DPSCs (or endothelial cells) in the previous study, it is known that MSCs generate a scalable yield of exosomes [33,34]. Exosomes consist of various cargoes, and proteins, especially, in the membrane and cytosol are used for exosome markers. In this study, we validated the exosome compositions including tetraspanins (CD63 and CD81), endosomal sorting complex required for transport (ESCRT) (TSG101 and ALIX), and membrane transport and fusion (ANXA5 and FLOT1) using an antibody array (Figure 1F) [35]. The spheroid morphology of exosomes was confirmed in SEM, and the size distribution (~100 nm) was consistent with the result of NTA (Figure 1G,H). Moreover, exosomes successfully internalized into recipient DPSCs within 48 h of exposure (Figure 1I,J).

The main appeal of our DPSC-Exo-based cell homing strategy lies in their ability to heal by self-congregating at exposure sites. Cell homing or migration, which is a prerequisite process during tissue regeneration, is defined as a directional cell movement in response to chemoattractant and is extremely important in both the development and maintenance of multicellular organisms. Recently, exosomes have been shown to play critical roles in cell migration due to the function of directional sensing, leader-follower behavior, cell adhesion, and extracellular matrix (ECM) degradation [36,37]. Specifically, exosomes contain cyclic adenosine monophosphate (cAMP) that can be actively synthesized and released to promote chemotaxis [38]. Sung and colleagues have reported that migrating cells leave stationary exosome trails to be used as paths for follower cells [39]. In this study, pHluo_M153R-CD63-positive exosome trails were observed behind cells, and following cells exhibited pathfinding behavior on the trails. In addition, cell adhesion molecules, such as integrins and fibronectin highly contained in exosomes, regulate cell migration speed [40]. Lastly, cell migration or invasion is highly achieved by proteolytic ECM degradation to allow for a cell path, and exosomes carry membrane-linked matrix metalloproteinases (MMPs) to promote the invasive motility of cells across ECM [41,42].

Exosomes or conditioned medium cultured from lineage-specific differentiated human bone marrow-derived stromal cells (HMSCs) and DPSCs have a great potential for odontogenesis. Hu and colleagues have shown that exosomes isolated under an odontogenic differentiation condition induced dramatic increases of odontogenic marker genes such as bone morphogenetic protein 9 (BMP9), alkaline phosphatase (ALP), Runt-related transcription factor 2 (RUNX2), and type I collagen [28,43]. In this study, we compared the effects of Exo-A and Exo-G on activities related to tissue regeneration. However, despite differences in miRNA content, there were almost no differences between Exo-G and Exo-A in terms of effects on DPSC proliferation, migration, and angiogenic differentiation. One possible reason for these findings is that proteins carried by both Exo-A and Exo-G were primarily responsible for the observed biologic activities. Moreover, we did not probe for the effects of exosomes on signal transduction pathways, which may have been differentially regulated by Exo-A and Exo-G miRNAs. These hypotheses will be tested in future studies.

Previous studies have reported that the conditioned medium and lysates of human DPSCs promoted angiogenesis by releasing angiogenic factors such as VEGF and monocyte chemotactic protein-1 (MCP-1) [44,45]. Regardless of a similar trend in the results, distinct levels of exosomal miRNA expression were identified in the comparison between Exo-G and Exo-A. Several mature miRNAs (ocu-miR-205-5p [46], ocu-let-7i-3p [47], ocu-miR-874-3p [48], and ocu-miR-29a-3p [47,49,50]) previously shown to be angiogenic were significantly over-represented in Exo-A versus Exo-G (Table 1). In contrast, ocu-miR-503-5p and ocu-miR-20b-5p, which were under-represented in Exo-A compared to Exo-G, may have allowed higher levels of CD40 [51] and hypoxia-inducible factor 1α (HIF-1α) expression [52] in Exo-A treated cultures, which could also lead to greater angiogenic stimulation. In order to further elucidate the difference between Exo-G and Exo-A, we plan to use a partial pulpotomy animal model, which can mimic clinical conditions in humans [53], and the in vivo effect of DPSC-Exos on cell homing and angiogenesis will be evaluated by immunohistochemistry analysis. In addition, the effect of exosomes derived from human DPSCs and the profile of exosomal miRNAs are needed to validate for clinical application.

miRNAs are small non-coding regulatory RNAs and play a role in post-transcriptional regulation of gene expression either by promoting messenger RNA (mRNA) degradation or by blocking the translation of transcribed sequences. Rather than targeting a specific gene, a single miRNA can regulate multiple pathways by modulating more than one Mrna [54]. miRNAs are secreted and transferred to target recipient cells via exosomes, thereby leading to functional cellular changes [55]. In this study, the profiles of miRNAs in DMSC-Exos strongly advocate for their potential in pulp regeneration. Both Exo-G and Exo-A contain mature (over 450), hairpin (over 250), and novel miRNAs (Figure 4, Appendix A). Among mature miRNA sequences, ocu-miR-199a-3p, ocu-miR-221-3p, ocu-miR-24-3p, and ocu-miR-21-5p were counted in over 100,000 total reads in both exosomes (Figure 4A–C). miR-199a-3p functions as a redundant network of the nitric oxide synthase (NOS)/nitric oxide (NO) pathway, thereby inducting endothelial tubulogenesis and cardiac regeneration [56,57]. The function of miR-221-3p is controversial in terms of angiogenesis. As an anti-angiogenic miRNA, it inhibits cell proliferation and migration in endothelial cells [58]. On the other hand, miR-221 is an essential regulatory node during angiogenesis [59]. Ocu-miR-24-3p exacted from adipose-derived mesenchymal stem cells can accelerate corneal epithelial migration in vitro and in vivo [60]. Lastly, miR-21-5p has a crucial role in regulating cell migration by ECM degradation and promoting angiogenesis via the extracellular signal-regulated kinase (ERK)/mitogen-activated protein kinase (MAPK) signal pathway [61,62]. Hence, targeting its associated miRNA(s) can serve as a novel therapeutic armamentarium for a given pathophysiologic condition including pulp damage. Once the profiled miRNA candidates are validated through in vivo and in silico modeling, the miRNA sequences can be loaded in engineered exosomes for clinical application, thereby effectively replacing DPSC cultures as a potential means of exosome production.

Most miRNAs are transcribed by RNA polymerase II to generate primary miRNAs (pri-miRNAs), which are then processed into hairpin precursor miRNAs (pre-miRNAs) via Drosha, a type III ribonuclease, resulting in mature miRNAs. In this study, we compared levels of hairpin miRNAs in DPSC-Exos from cells cultured under angiogenic or growth conditions (Exo-A versus Exo-G). A total of 291 hairpin miRNAs were found in DPSC-Exos. Among these, levels of 18 hairpin miRNAs were significantly different in the two groups (Table 2). Seven of these (ocu-miR-708, ocu-miR-134, ocu-miR-125b-2, ocu-miR-140, ocu-miR-29b-1, and ocu-miR-214) were present at significantly higher levels in Exo-A than in Exo-G, and five (ocu-miR-574, ocu-miR-503, ocu-miR-30a, ocu-miR-146a, and ocu-miR-671) were significantly lower in Exo-A versus Exo-G in both mature and hairpin miRNAs. Several of the miRNAs up-regulated in Exo-A have been strongly associated with tissue regeneration. For example, miR-708 was shown to protect against stress-induced apoptosis and promote myocardial regeneration in an animal study of injured hearts [63]. Sun and colleagues have also shown that miR-140-5p regulated DPSC proliferation and differentiation via the toll-like receptor 4 (TLR-4) [64]. Nevertheless, the functions of all listed miRNAs related to pulp regeneration, especially cell migration and angiogenesis, are still unknown. Therefore, further studies are needed to clarify their roles and associated regulatory pathways.

RT-PCR revealed a positive effect of DPSC-Exos on angiogenic marker expression only when angiogenic growth factors (fibroblastic growth factor B, VEGF, insulin-like growth factor-1, and epidermal growth factor) were present in the culture medium (Figure 3). Under these conditions, the expression of the angiogenic markers, VEGFA, FLT1, and PECAM1 was up-regulated up to four-fold by both Exo-G and Exo-A. In contrast, there was no effect of exosomes on cells cultured in basal medium lacking angiogenic growth factors, implying that exosomes may potentiate the effects of these factors. It is unlikely that these effects are attributable solely to exosomal miRNAs because, like MSC-derived exosomes, DPSC exosomes may contain peptide components including VEGF, extracellular matrix metalloproteinase inducer (EMMPRIN), and MMP-9, which play important roles in stimulating angiogenesis [65]. A proteomic analysis will be needed to fully characterize the protein content of DPSC-Exos.

## 4. Materials and Methods

### 4.1. Isolation of DPSCs

Two batches of the incisor pulp tissues obtained from New Zealand White rabbit cadavers (approximately 10 months old) were harvested to isolate DPSCs. Under sterile conditions, the dental pulp was minced into approximately 1 mm^3^ and plated in a culture dish. The tissue fragments and migrated cells were cultured in alpha minimum essential medium (α-MEM; Thermo Fisher Scientific, Waltham, MA, USA) supplemented with 10% fetal bovine serum (FBS; Thermo Fisher Scientific), 50 µg/mL L-ascorbic acid 2-phosphate trisodium salt (FUJIFILM Wako Chemicals, Richmond, VA, USA), 100 U/mL penicillin-streptomycin (Thermo Fisher Scientific), and 2.5 µg/mL amphotericin B (Sigma-Aldrich, St. Louis, MO, USA) in a hypoxic culture condition (5% O_2_/CO_2_ at 37 °C) [66]. Additionally, the cells were cultured in angiogenic differentiation induction media, EGM^TM^-2 endothelial cell growth medium (Lonza, Bend, OR, USA).

### 4.2. Characterization of DPSCs

Characterization of DPSCs (3rd–5th passage) was examined by multipotential differentiation. Angiogenic differentiation was evaluated using an endothelial tube formation assay (Cell Biolabs, San Diego, CA, USA) according to the manufacturer’s instructions. In brief, 1.5 × 10^4^ cells cultured in angiogenic induction medium for 10 days were seeded in a 16-well chamber slide (Thermo Fisher Scientific) coated with 50 µL extracellular matrix (ECM) gel. After 4 h, the cells were incubated with 1× Staining solution and imaged by an Olympus FV1000 confocal microscope (Olympus, Center Valley, PA, USA). For odontogenesis, 1.5 × 10^4^ cells were cultured with osteogenic differentiation medium (ScienCell, Carlsbad, CA, USA) in a 6-well plate, and Alizarin Red S staining (Sigma-Aldrich) was used to evaluate the cellular calcium deposit at 10 and 20 days. The cells fixed in 4% formaldehyde were stained with 40 mM Alizarin Red S for 30 min (min) and imaged by an Olympus BX60 microscope (Olympus).

### 4.3. Isolation of DPSC-Exos

Exosomes were isolated from rabbit DPSCs cultured under a growth (Exo-G) or angiogenic differentiation (Exo-A) condition. For Exo-A isolation, the cells were cultured in an angiogenic differentiation medium for 10 days. The induction medium was replaced with α-MEM growth medium containing 10% exosome-depleted FBS, and the conditioned medium (CM) was collected after 48 h. CM collected from non-inducted DPSCs was prepared for Exo-G isolation. DPSC-Exos were isolated by a precipitation method using ExoQuick-TC^TM^ (System Biosciences, Palo Alto, CA, USA). In brief, CM was centrifuged at 3000 × g for 15 min to remove cells and cell debris. The supernatant was mixed with an ExoQuick-TC^TM^ solution at 4 °C overnight. After centrifugation, the exosome pellet was resuspended in PBS and stored at −80 °C before use.

### 4.4. Nanoparticle Tracking Analyzer (NTA)

NTA was utilized to obtain the size distribution and concentration of DPSC-Exos using a NanoSight instrument (Malvern Panalytical, Westborough, MA, USA). Exosomes suspended in PBS were placed into a sonic bath at 30 °C for 10 min. The samples were then diluted for analysis with 0.1 µm-filtered PBS and injected into the laser viewing module via an automated syringe pump. The sample was illuminated using project-specific optical configuration settings according to the manufacturer’s protocol. Nanoparticles were tracked individually, counted, and sized via a translational diffusion coefficient. Triple samples were analyzed with the following parameters: syringe flow rate = 13, detection threshold = 3, track length = 10, and camera = 9–15. The data are reported in a number-weighted distribution. Latex beads of 100 nm diameter were used as a reference.

### 4.5. Exosome Antibody Array

The identification of DPSC-Exos was validated using an Exo-Check^TM^ exosome antibody array (System Biosciences). The array had eight positive markers (CD63, EpCAM: epithelial cell adhesion molecule, ANXA5: annexin A5, TSG101: tumor susceptibility gene 101, FLOT1: flotillin-1, ICAM: intercellular adhesion molecule 1, ALIX: programmed cell death 6 interacting protein, and CD81) and four controls including cis-golgi matrix protein (GM130) as a negative marker for cellular contamination during exosome isolation, two positive controls (+ ctrl), and a blank control. According to the manufacturer’s protocol, 50 µg DPSC-Exos suspended in PBS were mixed with 10% (*v*/*v*) lysis buffer and 1 µL labeling reagent. After 30 min incubation at room temperature (RT), the excess labeling reagent was removed using a column filter. A membrane was immersed in the mixture of labeled exosome lysate and 5 mL blocking buffer at 4 °C overnight on a shaker. After washing, the membrane was developed with WesternBright Sirius HRP substrate (Advansta, Menlo Park, CA, USA).

### 4.6. Scanning Electron Microscopy (SEM)

DPSC-Exos were fixed in 2.5% glutaraldehyde in 0.1 M sodium cacodylate and loaded on a clean silicon wafer (Ted Pella, Redding, CA, USA) on the top of a 37 °C hotplate. The wafer was covered with 100% hexamethyldisilazane (Ted Pella) for 10 min and then washed with PBS. After drying overnight, the samples were mounted on an aluminum stub (Ted Pella) with double-sided carbon tape and processed for an iridium sputter coating (Q150R-S, Quorum Technologies, Laughton, UK) to make the surface conductive. The final product was imaged in a high-resolution SEM (S-4800, Hitachi High-Tech, Ibaraki, Japan).

### 4.7. Exosome Labeling and Cellular Uptake

DPSC-Exos were labeled with a PKH67 green lipid membrane dye (Sigma-Aldrich) according to the manufacturer’s instructions. Briefly, 5 × 10^8^/mL exosomes suspended in 1 mL diluent C were mixed with 4 µL PKH67 and incubated for 5 min at room temperature. An equal volume of 5% bovine serum albumin was added to stop the labeling, and then the mixture was centrifuged with 110,000 g for 2 h at 4 °C. After washing with PBS and repeating centrifugation, the labeled exosomes were co-cultured with DPSCs for 48 h and imaged by an Olympus FV1000 confocal microscope (Olympus, Center Valley, PA, USA). Exosomes without PKH67 labeling were used for a negative control.

### 4.8. Cell Toxicity

DPSCs were seeded at a density of 1 × 10^4^ (200 µL) in 96-well plates. The next day, the cells were treated with various concentrations of DPSC-Exos (Exo-L: 5 × 10^7^/mL, Exo-M: 5 × 10^8^/mL, or Exo-H: 5 × 10^9^/mL) in a serum-free culture medium. Vehicle controls, 4.3%, 9.0%, and 16.3% (*v*/*v*) PBS, and no exosome treatment control, were prepared because the final products of exosomes were diluted in PBS. After 24 h, cytotoxicity was determined by CellTiter 96^®^Aqueous One Solution (Promega, Madison, WI, USA). One Solution Reagent (20 µL) was added into each well of the 96-well plates (100 µL medium), and the plates were incubated at 37 °C in a humidified incubator for 4 h. The absorbance was recorded at 490 nm using a 96-well plate reader (Infinite^®^ 200 PRO, TECAN, Männedorf, Switzerland).

### 4.9. Cell Proliferation

The cells were seeded at a density of 1 × 10^4^ (200 µL) in 48-well plates. After 2 h for cell adhesion, the media were replaced with 200 µL of treatment groups: controls (serum-free medium or serum-free medium with 10% exosomes-depleted FBS) and 5 × 10^7^/mL or 5 × 10^8^/mL DPSC-Exos (Exo-G or Exo-A). The media were changed every two days. At day 4, cell viability was evaluated using CellTiter 96^®^Aqueous One Solution (Promega).

### 4.10. Cell Migration

The cells (1 × 10^4^/100 µL) were added to the polycarbonate membrane inserts of 24-well Transwell^®^ plates with an 8 µm pore (Corning, Corning, NY, USA). In the reservoirs, 600 µL of serum-free medium or DPSC-Exos (5 × 10^7^/mL and 5 × 10^8^/mL Exo-G or Exo-A) were added. Following 48 h incubation, the inserts were swabbed with sterile cotton tips to remove non-migrated cells and washed twice in Hanks’ Balanced Salt Solution (HBSS). The inserts were stained with Calcein AM (1:1000 dilution; Thermo Fisher Scientific) and imaged by a confocal microscope. Separately, the migrated cells were digested in papain digestion buffer (1 mg/mL papain, 5 mM L-cysteine hydrochloride acid, 100 mM disodium hydrogen phosphate, 5 mM ethylenediaminetetraacetic acid salt) for 2 h and quantified using a Quant-iT^TM^ PicoGreen^®^ dsDNA Assay kit (Thermo Fisher Scientific). Fluorescence was measured on the plate reader at 480 nm excitation and 520 nm emission.

### 4.11. Angiogenic Differentiation

A total of six groups were prepared to evaluate the effect of DPSC-Exos on angiogenic differentiation: (1) basal angiogenic (bAM) differentiation medium without growth factors, (2) bAM + Exo-G, (3) bAM + Exo-A, (4) completed angiogenic (cAM) differentiation medium with growth factors, (5) cAM + Exo-G, and (6) cAM + Exo-A. DPSCs were seeded at a density of 3 × 10^5^ (2 mL) in 6-well plates, and the medium with or without 5 × 10^8^ DPSC-Exos was replaced every 2 days. After 7 days, the cells were trypsinized for gene expression analysis.

### 4.12. Real-Time Polymerase Chain Reaction (RT-PCR)

The cells were lysed to collect ribonucleic acid (RNA) using an RNAqueous^TM^ Total RNA Isolation kit (Thermo Fisher Scientific) for reverse transcription. A qualitative and quantitative assessment of RNA samples was performed with a Nanodrop^TM^ 2000 spectrophotometer (Thermo Fisher Scientific) prior to complementary deoxyribonucleic acid (cDNA) synthesis using a SuperScript^TM^ VILO^TM^ cDNA Synthesis kit (Thermo Fisher Scientific). The Taqman^TM^ gene expression assay was performed using a QuantStudio^TM^ 3 RT-PCR System (Applied Biosystems, Foster City, CA, USA) in a 10 µL PCR reaction mix containing 2 µL of cDNA template and an appropriate concentration of the Taqman^TM^ Advanced Master Mix and probes as recommended by the manufacturer. We used species-specific, premade probes (Thermo Fisher Scientific) labeled with a fluorescein (FAM) reporter and a minor groove binder (MGB) quencher both for housekeeping (GAPDH: Oc03823402_g1) and three target genes, vascular endothelial growth factor A (VEGFA: Oc03395999_m1), Fms-related tyrosine kinase 1 (FLT1: Oc06783656_s1), and platelet and endothelial cell adhesion molecule 1 (PECAM1: Oc06726473_m1). The relative changes in gene expression levels were calculated by the comparative C_T_ (∆∆C_T_) method [67].

### 4.13. Next-Generation Sequencing (NGS)

Two batches of rabbit DPSC-Exos were prepared for NGS analysis. Exosomal RNAs were extracted and quantified by a Bioanalyzer Small RNA Assay kit (Agilent, Santa Clara, CA, USA). NGS libraries were prepared and sequenced on a HiSeq^®^ Sequencing System (Illumina, San Diego, CA, USA) with 150 bp paired-end reads at an approximate depth of 10-15 million reads per sample in System Biosciences. Raw data were analyzed in the Bioinformatics Division of the Iowa Institute of Human Genetics (IIHG). To quantitate known small RNAs and to identify novel miRNAs, we used the Nextflow-based workflow nf-core/smrnaseq (version v1.1.0, https://github.com/nf-core/smrnaseq) (Accessed on 27 April 2022) as a small-RNA sequencing analysis pipeline. Rabbit (Oryctolagus cuniculus, build oryCun2) genome sequence (fasta) and annotation (gff) were downloaded from Ensembl (http://ftp.ensembl.org/pub/release-100) (Accessed on 27 April 2022). Sequences for mature and hairpin small RNAs were downloaded from miRBase (https://www.mirbase.org/ftp/CURRENT/) (Accessed on 27 April 2022). The adaptor protocol used was ‘qiaseq’, but for all other parameters, the default values were used. In brief, the workflow uses FastQC for assessing read quality, Trim Galore! for adapter trimming, seqcluster for read collapsing, and Bowtie1 for aligning reads to small RNA sequences. Read counts were imported into R and normalized using DESeq2. To determine differential expression, a model incorporating all the experimental factors was created, and Wald tests were used to compute statistical metrics. A small RNA was considered to have a statistically significant change in expression if the False Discovery Rate (FDR) was less than 10%. Results from the differential expression analysis were visualized using a volcano plot created with the ggplot2 package in R. To identify novel miRNAs, sequence read files (fastq) from samples in the same group were first concatenated together and then ran through the same nf-core/smrnaseq workflow as above with the miRDeep2 software used for novel miRNA identification. The results from miRDeep2 were filtered to include only potential novel miRNAs with miRDeep2 scores greater than or equal to 4 (higher quality).

### 4.14. Statistics

The scatter plots were expressed as the mean values with the standard deviation using GraphPad Prism (Version 8.1.2; San Diego, CA, USA). Data were compared by one-way ANOVA with the Tukey post-hoc test using SPSS Statistics software (Version 28; IBM, Armonk, NY, USA). Statistical significance was set at *p* < 0.05.

## 5. Conclusions

This study was designed to evaluate the effects of stem cell-derived exosomes on cell homing and angiogenic lineage-specific differentiation for pulp regeneration. Our results showed that DPSC-Exos (both Exo-G and Exo-A) significantly promoted cell proliferation, migration, and angiogenic differentiation although there was no superior effect of Exo-A on angiogenic differentiation compared to Exo-G in this in vitro study. In addition, we identified key miRNAs which regulate cell homing and angiogenesis. Therefore, our exosome-based cell homing and angiogenic differentiation strategy has a significant therapeutic potential for dental pulp regeneration.

## 6. Patents

The authors declare that there is potential intellectual property regarding the publication of this article.

## Figures and Tables

**Figure 1 ijms-24-00466-f001:**
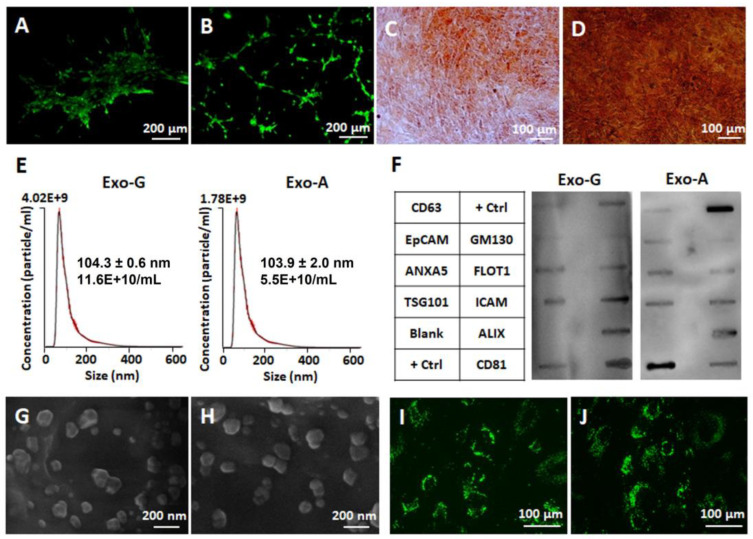
Characterization of rabbit dental pulp stem cells (DPSCs) and DPSC-derived exosomes (DPSC-Exos). (**A**,**B**) Tube formation as a marker of angiogenesis from DPSCs cultured in a regular medium (**A**) or angiogenic induction medium for 10 days (**B**). (**C**,**D**) Alizarin Red S staining as a marker of osteogenesis in osteogenic induction medium at 10 (**C**) and 20 (**D**) days. Exosomes were isolated from DPSCs cultured under a growth (Exo-G) or angiogenic differentiation condition (Exo-A). (**E**) Average size and concentration of Exo-G and Exo-A via Nanoparticle Tracking Analyzer (NTA). Red error bars indicate ± standard error of the mean (*n* = 3) (**F**) Exosome antibody array having eight positive markers (CD63, EpCAM: epithelial cell adhesion molecule, ANXA5: annexin A5, TSG101: tumor susceptibility gene 101, FLOT1: flotillin-1, ICAM: intercellular adhesion molecule 1, ALIX: programmed cell death 6 interacting protein, and CD81) and four controls including cis-golgi matrix protein (GM130) as a negative marker for cellular contamination during exosome isolation, two positive controls (+ Ctrl), and a blank control (Blank). (**G**,**H**) Scanning electron microscopy (SEM) images of Exo-G (**G**) and Exo-A (**H**). (**I**,**J**) Cellular uptake of Exo-G (**I**) and Exo-A (**J**) using PKH67 green fluorescence.

**Figure 2 ijms-24-00466-f002:**
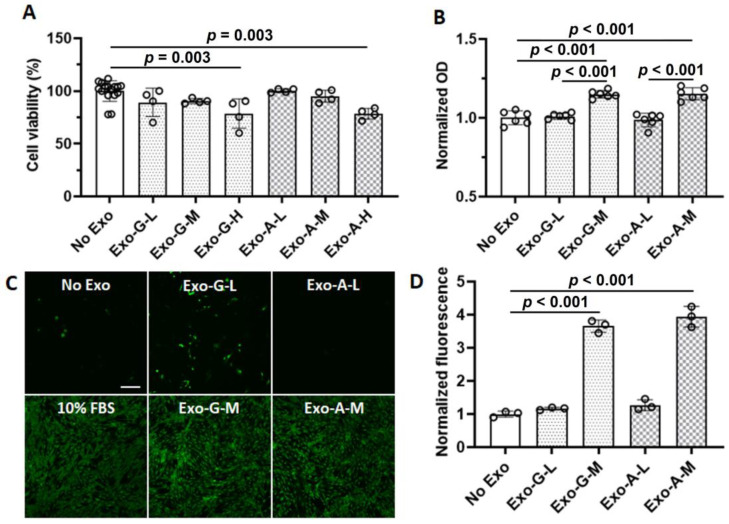
The effect of dental pulp stem cells-derived exosomes (DPSC-Exos) on cell toxicity, proliferation, and migration. Exosomes were isolated from DPSCs cultured under a growth (Exo-G) or angiogenic differentiation condition (Exo-A) and used at a concentration of 5 × 10^7^/mL (Exo-G-L or Exo-A-L), 5 × 10^8^/mL (Exo-G-M or Exo-A-M), or 5 × 10^9^/mL (Exo-G-H or Exo-A-H). (**A**) Cytotoxicity at 24 h (*n* = 4–16). (**B**) Cell proliferation at 4 days (*n* = 6). (**C**) Representative confocal images of migrated cells. FBS: fetal bovine serum. Green: Calcein AM. Scale bar: 200 µm. (**D**) Quantified fluorescence for cell migration (*n* = 3).

**Figure 3 ijms-24-00466-f003:**
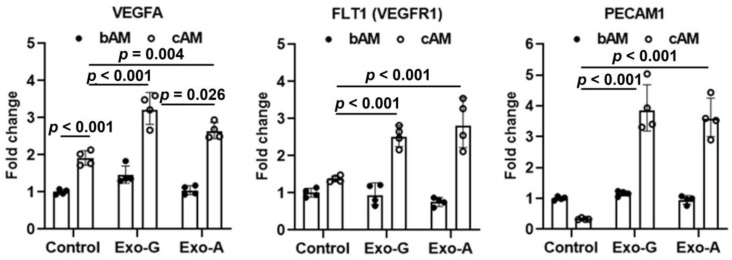
The effect dental pulp stem cells-derived exosomes (DPSC-Exos) on angiogenic differentiation. Exosomes (5 × 10^8^/mL) isolated from DPSCs cultured under a growth (Exo-G) or angiogenic differentiation condition (Exo-A) were treated in either basal or complete angiogenic induction medium (bAM or cAM) (*n* = 4). Control: no exosome treatment. VEGFA: vascular endothelial growth factor A, FLT1: Fms-related tyrosine kinase 1 (known as VEGF receptor 1), and PECAM1: platelet and endothelial cell adhesion molecule 1.

**Figure 4 ijms-24-00466-f004:**
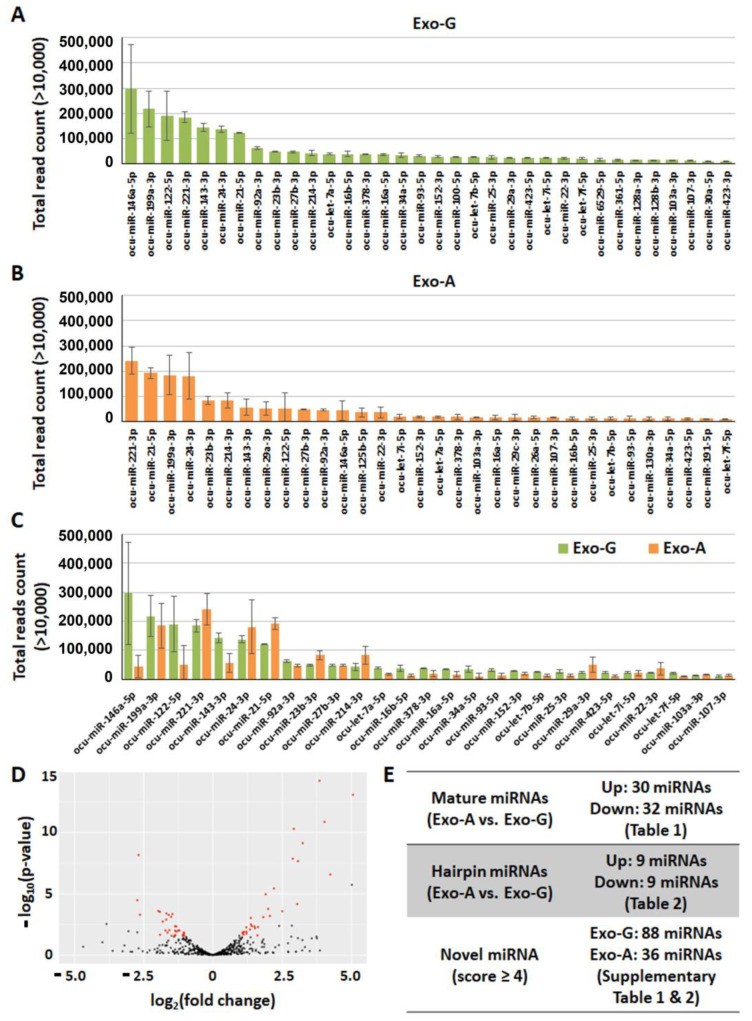
MicroRNA (miRNA) profiling of rabbit (Oryctolagus cuniculus: ocu) dental pulp stem cells-derived exosomes (DPSC-Exos) via next-generation sequencing (NGS). Exosomes were isolated from DPSCs cultured under growth (Exo-G) or angiogenic differentiation conditions (Exo-A). (**A**) Mature miRNAs with over 10,000 total read counts in Exo-G. (**B**) Mature miRNAs with over 10,000 total read counts in Exo-A. (**C**) Mature miRNAs with over 10,000 total read counts in both Exo-G and Exo-A. (**D**) Volcano plot of Exo-A versus Exo-G. Red: significantly up/down-expressed miRNAs. (**E**) The number of identified miRNAs.

**Table 1 ijms-24-00466-t001:** List of significantly up/down-expressed mature microRNAs (miRNAs): Exo-A versus Exo-G.

miRNA	log2(Fold Change)	*p*-Value	miRNA	log2(Fold Change)	*p*-Value
**ocu-miR-708-5p**	5.06	0.0000	**ocu-miR-146a-5p**	−2.73	0.0000
**ocu-miR-205-5p**	4.24	0.0000	**ocu-miR-503-5p**	−2.69	0.0000
**ocu-miR-708-3p**	4.03	0.0000	**ocu-miR-20b-5p**	−2.63	0.0005
**ocu-miR-885-5p**	3.85	0.0000	**ocu-miR-18a-3p**	−1.97	0.0003
**ocu-miR-24-2-5p**	3.24	0.0000	**ocu-miR-18a-5p**	−1.92	0.0003
**ocu-miR-29c-3p**	3.07	0.0000	**ocu-miR-122-5p**	−1.91	0.0223
**ocu-let-7i-3p**	3.04	0.0001	**ocu-miR-421-3p**	−1.81	0.0019
**ocu-miR-574-3p**	2.92	0.0000	**ocu-miR-350-5p**	−1.72	0.0099
**ocu-miR-874-3p**	2.88	0.0000	**ocu-miR-1307-3p**	−1.69	0.0013
**ocu-miR-134-5p**	2.50	0.0003	**ocu-miR-98-5p**	−1.68	0.0004
**ocu-miR-125b-5p**	2.20	0.0000	**ocu-miR-432-5p**	−1.64	0.0147
**ocu-miR-34c-3p**	2.06	0.0006	**ocu-miR-100-5p**	−1.60	0.0005
**ocu-miR-140-3p**	1.99	0.0002	**ocu-miR-34a-5p**	−1.58	0.0094
**ocu-miR-342-3p**	1.90	0.0000	**ocu-miR-16b-5p**	−1.51	0.0008
**ocu-miR-145-5p**	1.82	0.0008	**ocu-miR-20a-5p**	−1.50	0.0271
**ocu-miR-150-5p**	1.63	0.0053	**ocu-miR-30a-5p**	−1.46	0.0004
**ocu-miR-148a-5p**	1.61	0.0251	**ocu-miR-93-5p**	−1.38	0.0238
**ocu-miR-532-3p**	1.52	0.0065	**ocu-miR-6529-5p**	−1.37	0.0045
**ocu-miR-30b-5p**	1.50	0.0046	**ocu-miR-143-3p**	−1.37	0.0104
**ocu-miR-31-5p**	1.40	0.0044	**ocu-miR-652-3p**	−1.36	0.0241
**ocu-miR-214-5p**	1.37	0.0031	**ocu-miR-574-5p**	−1.33	0.0123
**ocu-miR-222-3p**	1.37	0.0010	**ocu-miR-7a-5p**	−1.32	0.0047
**ocu-miR-323a-3p**	1.34	0.0082	**ocu-miR-151-3p**	−1.26	0.0102
**ocu-miR-502a-3p**	1.21	0.0126	**ocu-miR-15b-3p**	−1.23	0.0097
**ocu-miR-130a-3p**	1.21	0.0185	**ocu-miR-671-5p**	−1.21	0.0101
**ocu-miR-12092-3p**	1.20	0.0055	**ocu-let-7f-5p**	−1.09	0.0268
**ocu-miR-29b-3p**	1.13	0.0128	**ocu-let-7a-5p**	−1.08	0.0220
**ocu-miR-29a-3p**	1.10	0.0216	**ocu-miR-423-5p**	−1.06	0.0259
**ocu-miR-335-5p**	1.07	0.0174	**ocu-miR-125b-3p**	−1.06	0.0146
**ocu-miR-181b-5p**	1.07	0.0143	**ocu-miR-361-5p**	−1.04	0.0240
			**ocu-miR-128a-3p**	−1.04	0.0161
			**ocu-miR-128b-3p**	−1.04	0.0161

**Table 2 ijms-24-00466-t002:** List of significantly up/down-expressed hairpin microRNAs (miRNAs): Exo-A versus Exo-G. * Significant expression of both mature and hairpin miRNAs.

miRNA	log2(Fold Change)	*p*-Value	miRNA	log2(Fold Change)	*p*-Value
**ocu-mir-708 ***	6.70	0.0000	**ocu-mir-574 ***	−3.05	0.0007
**ocu-mir-134 ***	3.06	0.0001	**ocu-mir-503 ***	−2.68	0.0000
**ocu-mir-125b-2 ***	2.65	0.0000	**ocu-mir-590**	−2.31	0.0133
**ocu-mir-140 ***	2.24	0.0000	**ocu-mir-30a ***	−1.94	0.0001
**ocu-mir-29b-2 ***	2.04	0.0006	**ocu-mir-146a ***	−1.93	0.0041
**ocu-mir-29b-1 ***	1.95	0.0011	**ocu-mir-671 ***	−1.77	0.0014
**ocu-mir-214 ***	1.54	0.0003	**ocu-mir-181b-2**	−1.49	0.0112
**ocu-mir-26a**	1.35	0.0013	**ocu-mir-152**	−1.38	0.0106
**ocu-mir-23b**	1.05	0.0132	**ocu-mir-155**	−1.13	0.0104

## Data Availability

Not applicable.

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
