# Peer review of "Exosome-Based Cell Homing and Angiogenic Differentiation for Dental Pulp Regeneration"

_ijms, 2022, doi:10.3390/ijms24010466_

Round 1
Reviewer 1 Report
The study entitled “Exosome-Based Cell Homing and Angiogenic Differentiation for Dental Pulp Regeneration” is an interesting domain of material science but article could not found as novel piece of work because the similar approach has already been reported in different journals. Moreover, the provided literature is very short and has typos errors throughout the manuscript.
The author should provide the brief overview such as Comparison of RCT, pulp revascularization and pulp regeneration via treatment procedures, root canal fillings, outcomes, complications, and limitations. of the approach and the significance of the study as well.
The author should provide the data of ultra-structure of Exosomes using Transmission electron and Scanning electron microscope.
3 The data didn’t mention the Controlled Release of Exosomes so author should evaluate the property and incorporate into the final version of the manuscript.
4 The author should justify the data of Cell Homing and angiogenic differentiation for Dental Pulp Regeneration using histological analysis and immunocytochemistry.
5 The study didn’t mention the data regarding the uptake of exosomes by cells.
6 Exosomes identification can do with specific biological marker such as tetraspanins proteins, exosomes biogenesis involved proteins, and endosomal sorting complex components that should be analyzed for the validation of vesicles.
7 The author characterized the exosomes by analyzing the presence of the exosome markers, CD63 and CD9, this will be easily detected by Western blotting.
8 Exosomes can promote regeneration via increase proliferation and migration ability of mesenchymal and epithelial cells through WNT, MAPK/ERK, TGFβ, and PI3K/AKT signaling pathways. The pathway should analyze in order to validate your data.
9 Exosomes derived from DPSCs promoted the odontogenic differentiation of DPSCs via transfer of miR-5100, miR-27a-5p, miR-652-3p. The important marker should be analyzed.
The discussion needs to be improved.
Reviewer 2 Report
In this article, authors proposed the use of exosomes to induce angiogenic differentiation and dental pulp regeneration.
I suggested some modifications.
Comments:
- Authors used for all study rabbit-derived cells. Why do use chose animal cells and not human cells? In your opinion same results could be obtained on human cell lines?
- In conclusion section no information about the better culture conditions to induce the exosome production (angiogenic differentiation or control).
- The exosome isolation method used in this study was not scalable. In order to use exosome in clinical practice, it is important to use a standardized and scalable method. Do you evaluated this aspect?
- Do you have any idea about the next step of this study? Do you want to test in vivo the exosomes?
Round 2
Reviewer 1 Report
The suggested points are addressed. The Manuscript is improved.
Reviewer 2 Report
Authors modified the manuscript as requested